# A role for gorilla APOBEC3G in shaping lentivirus evolution including transmission to humans

Yusuke Nakano[1], Keisuke Yamamoto[1,2], Mahoko Takahashi Ueda[3], Andrew Soper[1,2], Yoriyuki Konno[1,4,5], Izumi Kimura[1,4,6], Keiya Uriu[4,7], Ryuichi Kumata[1,4,8], Hirofumi Aso[1,4,6], Naoko Misawa[1], Shumpei Nagaoka[1,4,5], Soma Shimizu[1,6], Keito Mitsumune[1,6], Yusuke Kosugi[1,6], Guillermo Juarez-Fernandez[1,2], Jumpei Ito[1,4], So Nakagawa[3], Terumasa Ikeda[9,10,11], Yoshio Koyanagi[1,2,6], Reuben S. Harris[9,10], Kei Sato[1,4,6,7,12]*

1 Laboratory of Systems Virology, Institute for Frontier Life and Medical Sciences, Kyoto University, Kyoto, Japan, 2 Graduate School of Medicine, Kyoto University, Kyoto, Japan, 3 Department of Molecular Life Science, Tokai University School of Medicine, Kanagawa, Japan, 4 Division of Systems Virology, Department of Infectious Disease Control, International Research Center for Infectious Diseases, Institute of Medical Science, the University of Tokyo, Tokyo, Japan, 5 Graduate School of Biostudies, Kyoto University, Kyoto, Japan, 6 Graduate School of Pharmaceutical Sciences, Kyoto University, Kyoto, Japan, 7 Graduate School of Medicine, the University of Tokyo, Tokyo, Japan, 8 Faculty of Science, Kyoto University, Kyoto, Japan, 9 Department of Biochemistry, Molecular Biology and Biophysics, Masonic Cancer Center, Institute for Molecular Virology, Center for Genome Engineering, University of Minnesota, Minneapolis, Minnesota, United States of America, 10 Howard Hughes Medical Institute, University of Minnesota, Minneapolis, Minnesota, United States of America, 11 Division of Molecular Virology and Genetics, Joint Research Center for Human Retrovirus Infection, Kumamoto University, Kumamoto, Japan, 12 CREST, Japan Science and Technology Agency, Saitama, Japan

* ksato@ims.u-tokyo.ac.jp

## Abstract

The APOBEC3 deaminases are potent inhibitors of virus replication and barriers to cross-species transmission. For simian immunodeficiency virus (SIV) to transmit to a new primate host, as happened multiple times to seed the ongoing HIV-1 epidemic, the viral infectivity factor (Vif) must be capable of neutralizing the APOBEC3 enzymes of the new host. Although much is known about current interactions of HIV-1 Vif and human APOBEC3s, the evolutionary changes in SIV Vif required for transmission from chimpanzees to gorillas and ultimately to humans are poorly understood. Here, we demonstrate that gorilla APOBEC3G is a factor with the potential to hamper SIV transmission from chimpanzees to gorillas. Gain-of-function experiments using SIVcpz*Ptt* Vif revealed that this barrier could be overcome by a single Vif acidic amino acid substitution (M16E). Moreover, degradation of gorilla APOBEC3F is induced by Vif through a mechanism that is distinct from that of human APOBEC3F. Thus, our findings identify virus adaptations in gorillas that preceded and may have facilitated transmission to humans.

**Data Availability Statement:** The data is available at GitHub (https://github.com/TheSatoLab).

**Funding:** This study was supported in part by AMED J-PRIDE 19fm0208006 (to K.S.); AMED

Research Program on HIV/AIDS 19fk0410019 (to
K.S.) and 19fk0410014 (to Y. Koyanagi and K.S.);
AMED Research Program on Emerging and Re-
emerging Infectious Diseases 20fk0108146 (to K.
S.), 19fk010817 (to S. Nakagawa and K.S.), and
20fk0108270 (to Y. Koyanagi and K.S.); JST
CREST (to K.S.); KAKENHI Grant-in-Aid for
Scientific Research B 18H02662 (to K.S.),
KAKENHI Grant-in-Aid for Scientific Research on
Innovative Areas 16H06429 (to S. Nakagawa and
K.S.), 16K21723 (to S. Nakagawa and K.S.),
17H05823 (to S. Nakagawa), 17H05813 (to K.S.),
19H04843 (to S. Nakagawa), and 19H04826 (to K.
S.), KAKENHI Early-Career Scientists 20K15767 (to
J.I.), and Fund for the Promotion of Joint
International Research (Fostering Joint
International Research) 18KK0447 (to K.S.); JSPS
Research Fellow DC1 19J22914 (to Y. Konno),
DC1 19J20488 (to I.K.), DC1 19J22802 (to S.
Nagaoka), 20J23299 (to H.A.), and PD 19J01713
(to J.I.); JSPS Leading Initiative for Excellent Young
Researchers (LEADER) (to T.I.); Takeda Science
Foundation (to K.S.); ONO Medical Research
Foundation (to K.S.); Ichiro Kanehara Foundation
(to K.S.); Lotte Foundation (to K.S.); Mochida
Memorial Foundation for Medical and
Pharmaceutical Research (to K.S.); Daiichi Sankyo
Foundation of Life Science (to K.S.); Sumitomo
Foundation (to K.S.); Uehara Foundation (to K.S.);
Joint Research Project of the Institute of Medical
Science, the University of Tokyo 3057 (to Y.N.);
Joint Usage/Research Center program of Institute
for Frontier Life and Medical Sciences, Kyoto
University (to S. Nakagawa and K.S.); and JSPS
Core-to-Core program (A. Advanced Research
Networks) (to Y. Koyanagi, R.S.H. and K.S.);
International Joint Research Project of the Institute
of Medical Science, the University of Tokyo 2019-
K3003 (to T.I., R.S.H. and K.S.); and NIAID R37
AI064046 (to R.S.H.). R.S.H. is the Margaret
Harvey Schering Land Grant Chair for Cancer
Research, a Distinguished McKnight University
Professor, and an Investigator of the Howard
Hughes Medical Institute. The funders had no role
in study design, data collection and analysis,
decision to publish, or preparation of the
manuscript.

**Competing interests:** I have read the journal's
policy and the authors of this manuscript have the
following competing interests: R.S.H. is a co-
founder, shareholder and consultant of ApoGen
Biotechnologies.

## Author summary

Humans are exposed continuously to a menace of viral diseases such as Ebola virus and
coronaviruses. Such emerging/re-emerging viral outbreaks can be triggered by cross-spe-
cies viral transmission from wild animals to humans. HIV-1, the causative agent of AIDS,
most likely originated from related precursors found in chimpanzees and gorillas
(SIVcpz*Ptt* or SIVgor), approximately 100 years ago. Additionally, SIVgor most likely
emerged through the cross-species jump of SIVcpz*Ptt* from chimpanzees to gorillas. How-
ever, it remains unclear how primate lentiviruses successfully transmitted among different
species. To limit cross-species lentiviral transmission, cellular "restriction factors", includ-
ing tetherin, SAMHD1, and APOBEC3 proteins potentially inhibit lentiviral replication.
In contrast, primate lentiviruses have evolutionary acquired their own "arms" to antago-
nize the antiviral effect of restriction factors. Here we show that gorilla APOBEC3G
potentially plays a role in inhibiting SIVcpz*Ptt* replication. To our knowledge, this is the
first report suggesting that a great ape APOBEC3 protein can potentially restrict the cross-
species transmission of great ape lentiviruses and how lentiviruses overcame this species
barrier.

## Introduction

Lentiviruses have been identified in great apes including humans (*Homo sapiens*), central
chimpanzees (*Pan troglodytes troglodytes*), eastern chimpanzees (*Pan troglodytes schwein-
furthii*), and gorillas (*Gorilla gorilla gorilla*). The lentiviruses isolated from these host species
are designated HIV [1], SIVcpz*Ptt* [simian immunodeficiency virus (SIV) in central chimpan-
zee] [2, 3], SIVcpz*Pts* (SIV in eastern chimpanzee) [4], and SIVgor (SIV in gorilla) [5]. HIV
type 1 (HIV-1) is classified into four groups, M (major), N (non-M-non-O), O (outlier) and P
[reviewed in [6]], and a molecular phylogenetic analysis indicates that HIV-1 groups M and N
originated from SIVcpz*Ptt* [3].

  SIVgor was first discovered from the fecal samples of wild gorillas, which were obtained in
remote forest regions in Cameroon in 2007 [5]. A subsequent study revealed that SIVgor is
phylogenetically related to HIV-1 groups O and P [7], suggesting that SIVgor is the ancestral
virus of these HIV-1 groups in the human population. Moreover, a phylogenetic analysis
deduced that SIVgor most likely emerged from the leap of SIVcpz*Ptt* from chimpanzees to
gorillas [8]. Thus, the HIV-1 pandemic was "seeded" by multiple cross-species viral transmis-
sion events from both chimpanzees and gorillas. The factors that may have contributed to
these ape-to-human cross-species transmission events and the broader spread of HIV-1 group
M in comparison to the other HIV-1 groups have been the topics of numerous studies
(reviewed in [6]). In contrast, relatively little is known about the virus-host interactions at play
in the other critical cross-species transmission events that preceded the emergence of HIV-1,
specifically chimpanzee-to-gorilla cross-species transmission events [9].

  Several host-encoded restriction factors, including SAMHD1, SERINC5, tetherin and APO-
BEC3 (A3) proteins, have the potential to limit HIV-1 infection (reviewed in [10, 11]).
SAMHD1 restricts lentiviral reverse transcription by impairing dNTP supply, but is degraded
and neutralized by the lentiviral Vpr or Vpx proteins [12]. SERINC5 is incorporated into
released virions and prevents viral fusion, but is antagonized by the lentiviral Nef protein [13].
Importantly, SERINC5 is neutralized efficiently by all tested Nef proteins of great ape lentivi-
ruses in a species-independent manner [13], whereas the Vpr proteins of great ape lentiviruses
do not neutralize SAMHD1 [12]. Therefore, these two restriction proteins are unlikely to have

been factors that limited the cross-species transmissions of SIVcpz and SIVgor in great apes including humans.

One of the well-studied host factors that potently restricts cross-species transmission of great ape lentiviruses is tetherin. The Vpu protein of pandemic HIV-1 group M down-modulates and antagonizes human tetherin [14, 15]. The Vpu protein of SIVcpz*Ptt*, the ancestral virus of HIV-1 group M [3], is incapable of counteracting human tetherin [16], implying that human tetherin acts as barrier restricting cross-species lentiviral transmission from chimpanzees to humans, and that acquiring anti-human tetherin activity was likely to have been essential for the successful cross-species jump (reviewed in [9, 17]).

Another group of well-understood restriction factors is the A3 family of single-stranded DNA cytosine deaminases. Most great apes encode seven A3 proteins [reviewed in [18]]. At least two, A3F and A3G, are packaged into nascent viral particles and suppress viral infectivity through inserting G-to-A mutations in the viral genome (reviewed in [19]). To counteract A3-mediated restriction action, the viral infectivity factor (Vif), an accessory protein of lentiviruses, recruits a cellular E3 ubiquitin ligase complex and degrades the host A3 proteins via a ubiquitin-proteasome-dependent pathway (reviewed in [19]). Because Vif-mediated counteraction of antiviral A3 is largely species-specific, it is not surprising that lentiviral *vif* and mammalian *A3* genes seem to have co-evolved [reviewed in [18]]. For instance, it has been reported that Old World monkey A3G proteins contribute to restricting lentiviral transmission (reviewed in [20]); specifically, the Vif protein of SIV in vervet monkey can neutralize the A3G protein of its natural host (vervet monkey) but not those of mustached guenon and De Brazza's monkey [21, 22].

The molecular interactions between HIV-1 group M Vif and human A3 proteins are well established (reviewed in [19]). Also, the functional and evolutionary relationships between SIVcpz*Ptt* Vif and human A3 proteins have been investigated [23–26]. However, the evolutionary adaptations of great ape lentiviruses to gorillas that resulted in the emergence of SIVgor from SIVcpz*Ptt* have not been fully investigated. Importantly, previous studies have demonstrated that SIVgor Vif can neutralize gorilla A3G (gA3G) but SIVcpz*Ptt* Vif cannot [7, 23]. However, the functional and evolutionary association of the Vif proteins of SIVcpz*Ptt* and SIVgor with their host A3 proteins remains unclear. Through molecular, phylogenetic, virological, and structural approaches, here we investigate the antiviral effects of gorilla A3s and their antagonization by SIVcpz*Ptt* and SIVgor Vif proteins. This work provides an evolutionary framework for better understanding interactions between great ape lentiviruses and great ape A3 proteins that preceded the emergence of HIV-1 groups O and P in humans.

## Results

### Gorilla A3G is resistant to SIVcpz*Ptt* Vif

We first constructed a phylogenetic tree of the *vif* genes of great ape lentiviruses (**Fig 1A**). Consistent with previous reports, HIV-1M and HIV-1N originated from SIVcpz*Ptt* [3], while HIV-1O and HIV-1P are more closely related to SIVgor [7] (**Fig 1A and 1B**). It should be also noted that HIV-1M, a progeny of SIVcpz*Ptt*, have infected more than 70 million people worldwide, while HIV-1O, a progeny of SIVgor, has mainly endemic in Western and Central Africa and has infected ~100,000 individuals (reviewed in [6, 9]) (**Fig 1B**). Only small numbers of patients infected with the other two group viruses, HIV-1N and HIV-1P, have been reported [7] (**Fig 1B**).

To address the possibility that great ape A3G can be a factor restricting cross-species transmission of great ape lentiviruses, we set out to analyze the antiviral activity of great ape A3G. We co-transfected the expression plasmids of great ape A3G (from humans, chimpanzees and

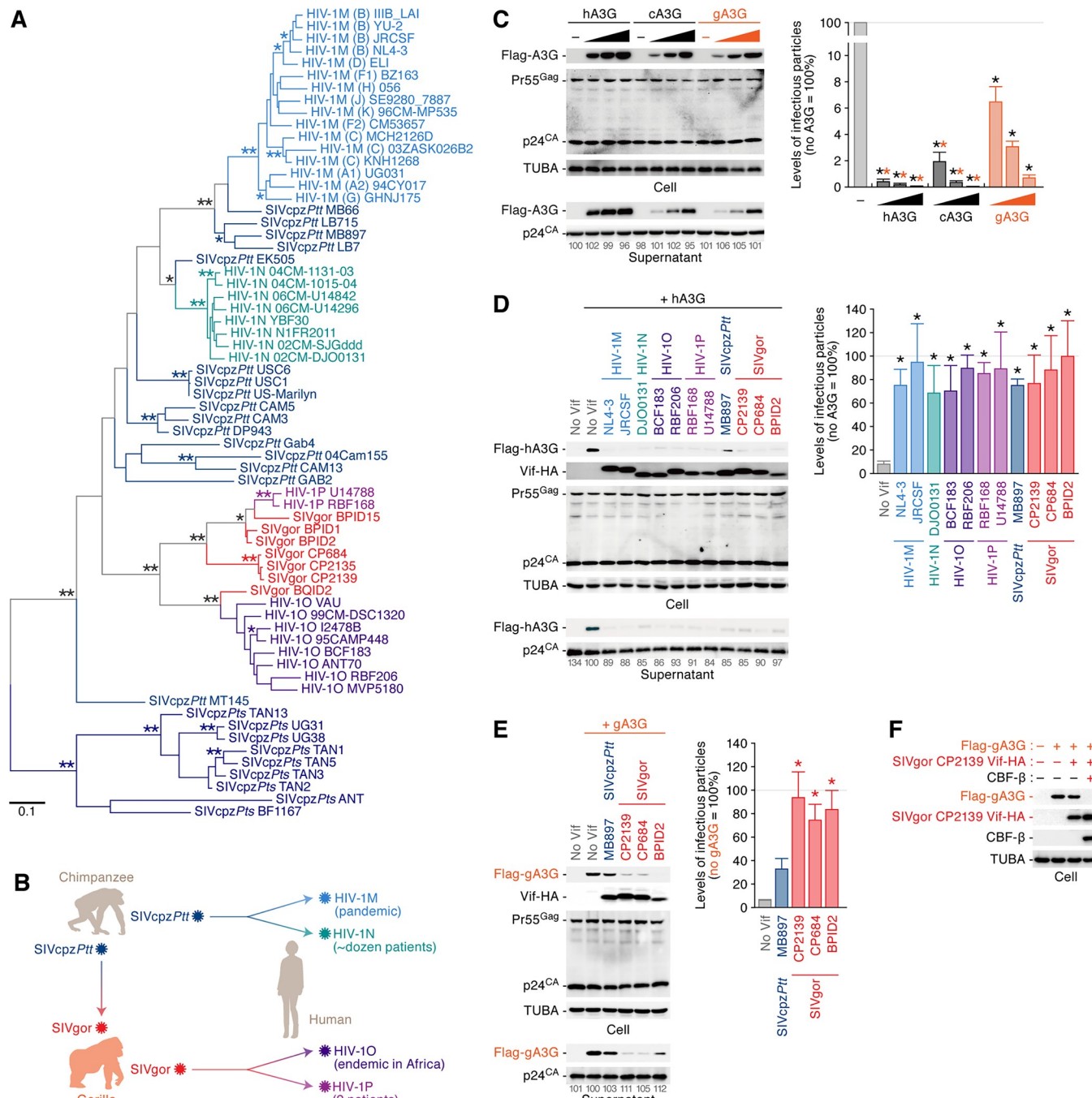

**Fig 1. Failure of neutralizing antiviral effect of gA3G by SIVcpzPtt Vif.** (A) A phylogenetic tree of the *vif* gene of great ape lentiviruses. The *vif* sequences were extracted from Los Alamos HIV sequence database (https://www.hiv.lanl.gov/components/sequence/HIV/search/search.html) and the phylogenetic tree was constructed as described in Materials and Methods. The bootstrap values are indicated as follows: *, >80%; **, >95%. A scale bar indicates 0.1 nucleotide substitutions per site. (B) A scheme of cross-species lentiviral transmission in great apes. (C) Antiviral activity of great ape A3G. HEK293T cells were co-transfected with pNL-3Δ*vif* (500 ng) and the different amounts of expression plasmids for great ape A3G (0, 50, 100, and 200 ng; the plasmid amount was normalized by empty vector). Cells and supernatants were harvested at two days post-transfection and were used for Western blotting (left) and TZM-bl assay (right). For Western blotting, the band intensity of viral p24 was quantified and the intensity value of the leftmost lane was set to 100%. For TZM-bl assay, the infectivity value without A3G was set to 100%. (D and E) Counteracting ability of great ape lentiviral Vif against great ape A3G. HEK293T cells were co-transfected with pNL-3Δ*vif* (500 ng) and the expression plasmids for hA3G (D; 10 ng) or gA3G (E; 50 ng) and the Vif of indicated viral strain (500 ng). In C-E, cells and supernatants were harvested at two days post-transfection and were used for Western blotting (left) and TZM-bl assay (right). For Western blotting, the band intensity of viral p24 was quantified and the intensity value of the A3G expressing cells without Vif (second from the left) was set to 100%. For TZM-bl assay, the percentage of the value without A3G are shown. The mean values of three independent experiments ± SEM are shown, and statistically significant

differences ($P < 0.05$) versus "no A3G (C, black asterisks)", "gA3G (C, orange asterisks)", or "no Vif" (D and E, asterisks) are shown. (F) CBF-β-dependent degradation of gA3G by SIVgor Vif. *CBFB* KD HEK293 cells were co-transfected with the expression plasmids for gA3G (50 ng), SIVgor Vif (500 ng) and CBF-β (400 ng). Cells were harvested at two days post-transfection and were used for Western blotting. For Western blotting, the input of cell lysate was standardized to TUBA, and representative results are shown.

gorillas) with an infectious molecular clone (IMC) of *vif*-deleted HIV-1. All great ape A3Gs exhibited dose-dependent antiviral effects, but gA3G was less antiviral than human A3G (hA3G) and chimpanzee A3G (cA3G) (**Fig 1C**). Additionally, although other gorilla A3 proteins such as A3D, A3F and A3H exhibited antiviral effects, these activities were relatively lower than the counterparts of human and chimpanzee (**S1 Fig**). Comparing the antiviral effect of gorilla A3 proteins, our data suggest that gA3G is a relatively more robust antiviral factor than other gorilla A3 proteins.

To assess the ability of lentiviral Vif to counteract host A3G, the expression plasmid for hA3G was co-transfected together with the Vif expression plasmids and an IMC of *vif*-deleted HIV-1. As shown in **Fig 1D**, all lentiviral Vifs including HIV-1MNOP, SIVcpz*Ptt* and SIVgor induced the degradation of hA3G and thus impaired its incorporation into the released virions. Additionally, viral infectivity was rescued by all lentiviral Vifs tested in this study (**Fig 1D**, **right**). These results show that viral Vif proteins from all tested great ape lentiviruses are sufficient to overcome the restriction mediated by hA3G, and thus that hA3G was unlikely to restrict transmission of SIVcpz or SIVgor to humans. In sharp contrast, gA3G could not be neutralized efficiently by SIVcpz*Ptt* Vif (**Fig 1E**), which is consistent with previous reports [7, 23]. We further revealed that SIVgor Vif-mediated degradation of gA3G requires CBF-β, a cofactor of HIV-1 Vif for hA3G degradation [27] (**Fig 1F**). These findings strongly suggest that gA3G may have helped to prevent cross-species transmissions of SIVcpz*Ptt* from chimpanzees to gorillas.

## M16E enables SIVcpz*Ptt* Vif to efficiently degrade gorilla A3G

Previous studies showed that the amino acid at position 129 of gA3G determines the sensitivity to SIVgor Vif-mediated degradation [7, 23]. Interestingly, only gA3G harbors a glutamine (Q) at position 129, while hA3G and cA3G possess proline (P) at this position [7, 23, 28]. However, because of the low sequence similarity on the *vif* genes of SIVcpz*Ptt* and SIVgor (the p-distance between SIVcpz*Ptt* Vif and SIVgor Vif is 0.373 ± 0.013), the responsible residue(s) in Vif determining the ability to counteract gA3G have not been identified. To investigate how SIVgor acquired the ability to counteract gA3G, we next performed gain-of-function experiments based on SIVcpz*Ptt* Vif (strain MB897). We first generated four chimeric Vif mutants of SIVcpz*Ptt* MB897 and SIVgor CP2139 (chimeras A-D; **Fig 2A** and **S2 Fig**) and evaluated their ability to counteract gA3G-mediated antiviral activity using cell-based single-round infection assays. As shown in **Fig 2B**, SIVgor CP2139 Vif as well as chimera A Vif overcame the gA3G-mediated antiviral effect, suggesting that the N-terminal region of SIVgor is responsible for the gA3G neutralization. We then generated five additional mutants (chimeras A1-A5) and performed cell-based co-transfection experiments. We found that only chimera A1 is able to induce the degradation of gA3G (**Fig 2C**, **left**) and therefore to rescue the infectivity of released viruses (**Fig 2C**, **right**). Only three amino acid differences occur in this region (**S2 Fig**). Individual mutants were analyzed, and only M16E, but not the K6Q or D14P mutations, conferred the ability to counteract gA3G (**Fig 2D**). To analyze the effect of the M16E mutation on viral spread, we constructed IMCs of SIVcpz*Ptt* MB897 expressing mutated (M16E) or no (E2X) Vif. In the absence of A3s, infectivity of the M16E and E2X (*vif*-deleted) variants was comparable to that of wild-type (WT) virus (**Fig 2E**). We used these viruses to infect human A3-null

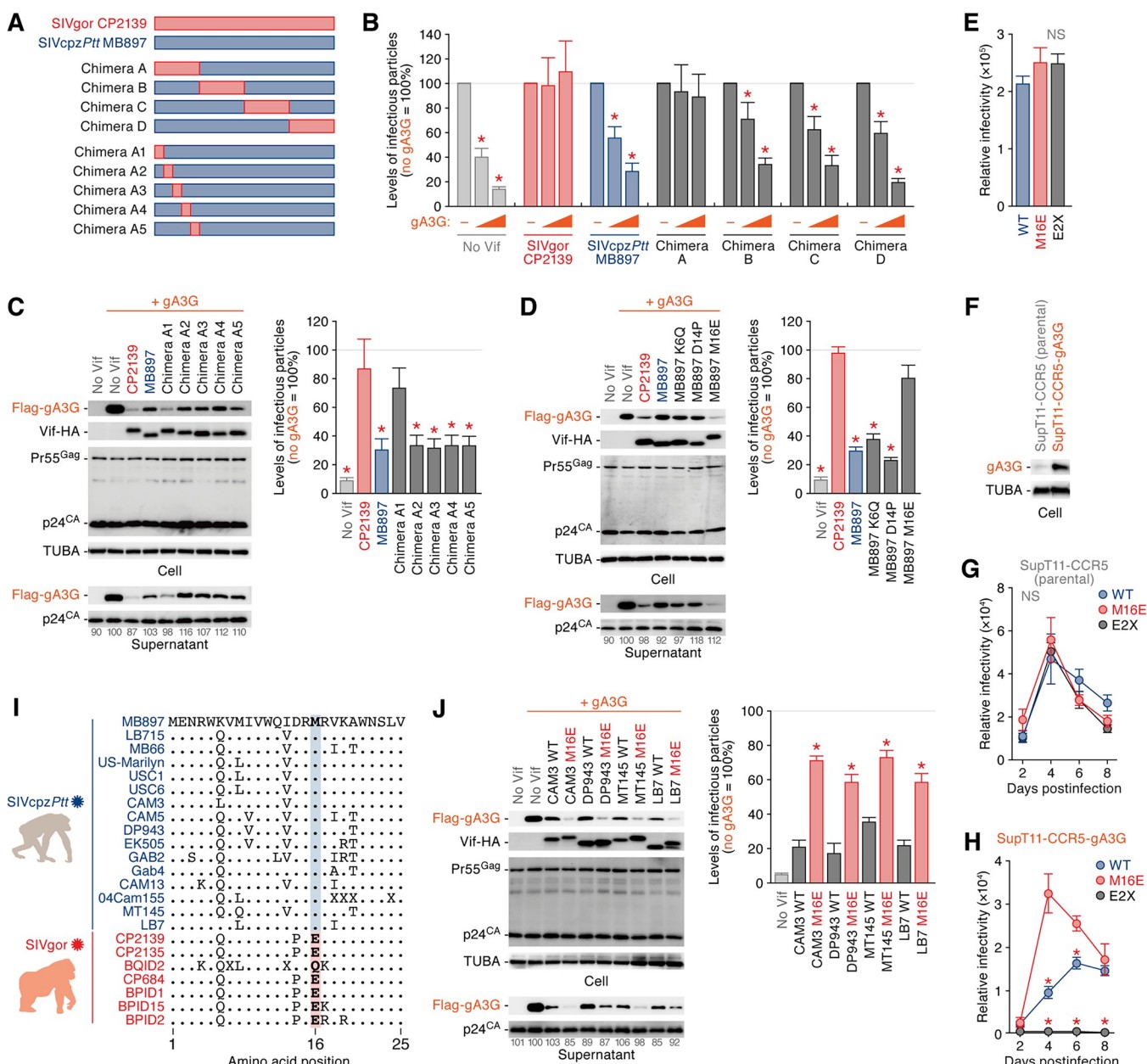

**Fig 2. Gain-of-function experiments of SIVcpz*Ptt* Vif to counteract gA3G.** (A) A scheme of SIVcpz*Ptt* MB897 Vif derivatives used in this study. (B-D) Determination of the amino acid residue of SIV Vif that is responsible to neutralize gA3G. HEK293T cells were co-transfected with pNL4-3Δ*vif* (500 ng) and the expression plasmids for gA3G (10 or 50 ng; the plasmid amount was normalized by empty vector) and the indicated Vif derivatives (500 ng). (E) Infectivity of the infectious viruses of SIVcpz*Ptt* MB897 WT, M16E and E2X (*vif* deleted) derivatives. These viruses were prepared as described in Materials and Methods and the infectivity was measured by using TZM-bl cells. The mean values of three independent experiments ± SEM are shown. (F-H) (F) Expression of gA3G in SupT11-CCR5 cells. The SupT11-CCR5 cells stably expressing gA3G (SupT11-CCR5-gA3G) was prepared as described in Materials and Methods. For Western blotting, the input of cell lysate was standardized to TUBA, and representative results are shown. (G and H) Multi-round replication assay of SIVcpz*Ptt*. The infectious viruses of SIVcpz*Ptt* MB897 WT, M16E and E2X (*vif*-deleted) derivatives were inoculated into parental SupT11-CCR5 cells (G) or the SupT11-CCR5-gA3G (H) at MOI 0.1. (Left) Culture supernatant was routinely harvested and the amount of infectious viruses was measured by using TZM-bl cells. (I) Conservation of residue 16 in the Vif proteins of all SIVcpz*Ptt* and SIVgor strains reported. Note that "X" is an undefined amino acid because of nucleotide ambiguity. (J) Importance of M16E substitution in other SIVcpz*Ptt* strains to counteract gA3G. HEK293T cells were co-transfected with pNL4-3Δ*vif* (500 ng) and the expression plasmids for gA3G (50 ng) and the indicated Vif strains and derivatives (500 ng). In B-D and J, cells and supernatants were harvested at two days post-transfection and were used for Western blotting and TZM-bl assay. For Western blotting, the input of cell lysate was standardized to TUBA, and representative results are shown. The band intensity of viral p24 was quantified and the intensity value of the gA3G expressing cells without Vif (second from the left) was set to 100%. For TZM-bl assay, the percentage of the value without gA3G is shown. The mean values of nine independent experiments ± SEM are shown. Statistically significant differences ($P < 0.05$) versus "CP2139" (B-D), "M16E" (G and H) or "WT" (J) are shown by red asterisks. NS, no statistical significance.

SupT11-CCR5 cells and SupT11-CCR5 cells stably expressing gA3G (SupT11-CCR5-gA3G) (**Fig 2F**). In parental SupT11-CCR5 cells, these three viruses replicated with similar kinetics (**Fig 2G**). In SupT11-CCR5-gA3G cells, however, the growth kinetics of the M16E mutant was significantly higher than those of WT and the E2X mutant (**Fig 2H**). These findings show that Vif position 16 is an important determinant of gA3G counteraction.

As shown in **Fig 2I**, Vif M16 is highly conserved among different SIVcpz*Ptt* strains, whereas most of SIVgor strains have a glutamic acid (E) at this position. Although a strain of SIVgor, BQID2, possessed a Q at position 16 (**Fig 2I**), we notified that the sequence quality of this strain is low; 7 out of the 597 *vif*-encoding nucleotides were ambiguous (e.g., R or W), and the full-length sequence of this strain (9,241 nucleotides; accession no. KP004991) contained 53 ambiguous nucleotides. These insights imply that a Q at position 16 of BQID2 Vif may due to its low quality sequence.

To investigate whether M-to-E substitution confers broad gA3G neutralization activity to SIVcpz*Ptt* Vif to counteract gA3G, we constructed the M16E mutants of four additional SIVcpz*Ptt* strains (i.e. CAM3, DP943, MT145 and LB7). Although the WT Vif proteins of these SIVcpz*Ptt* strains already weakly counteract gA3G, all M16E mutants tested in this experiment acquired the ability to efficiently trigger the degradation of gA3G (**Fig 2J**). These findings indicate that the acquisition of gA3G degradation activity through the M16E substitution is not limited to the Vif protein of SIVcpz*Ptt* strain MB897 but generally shared across different SIVcpz*Ptt* Vif proteins.

To assess whether the E16M reversion of SIVgor Vif renders it unable to counteract gA3G, we prepared the SIVgor CP2139 Vif E16M mutant. As shown in **S3 Fig**, the ability of the E16M mutant to degrade gA3G seems to be lower than that of parental SIVgor Vif. However, the level of gA3G in virions and viral infectivity of the E16M mutant were comparable to those of parental SIVgor Vif (**S3 Fig**). Although its mechanism(s) remain unclear, these data suggest that the E16M mutation of SIVgor Vif does not revert its ability to degrade gA3G.

## Acidic residue at position 16 is required to counteract gorilla A3G

We next assessed the side-chain properties of residue 16 of SIVcpz*Ptt* Vif that are required to counteract gA3G. To address this, we constructed three additional mutants (M16A, M16D and M16Q). In addition to the M16E mutant, the M16D mutant of MB897 Vif counteracted the gA3G-mediated antiviral effect, while the M16A and M16Q mutants did not (**Fig 3A**). These results suggest that an acidic residue at position 16 of SIVcpz Vif is necessary for counteracting gA3G.

To further investigate the effects of an acidic residue at position 16 on Vif structure and function, we used the crystal structure of HIV-1M Vif [29] to construct the structural homology models of SIVgor CP2139 Vif, SIVcpz*Ptt* MB897 Vif and key derivatives (i.e. M16E, M16A, M16D, and M16Q). As shown in **Fig 3B**, the structure models indicate that the two basic residues at positions 15 and 19, which are in the vicinity of residue 16, are exposed on the surface and adjacent to each other. We then compared the structure models of SIVgor CP2139 Vif and SIVcpz*Ptt* MB897 Vif derivatives to that of SIVcpz*Ptt* MB897 Vif and found that the topology of the residues at positions 15, 16, and 19 are clearly different. Particularly, the topology of the two basic residues at positions 15 and 19 in the Vif proteins that are capable of counteracting gA3G was changed (i.e., SIVgor CP2139, SIVcpz*Ptt* MB897 Vif M16E and M16D; **Fig 3C**). Furthermore, the two basic residues at positions 15 and 19 are located on the same face of the alpha-helix, and the structural locations of these two basic residues are changed in the Vif proteins that are competent to counteract gA3G (SIVgor CP2139, SIVcpz*Ptt* MB897 Vif M16E and M16D) (**Fig 3D**). These findings suggest that changing topology of these two basic residues may be crucial for gaining the ability to counteract gA3G.

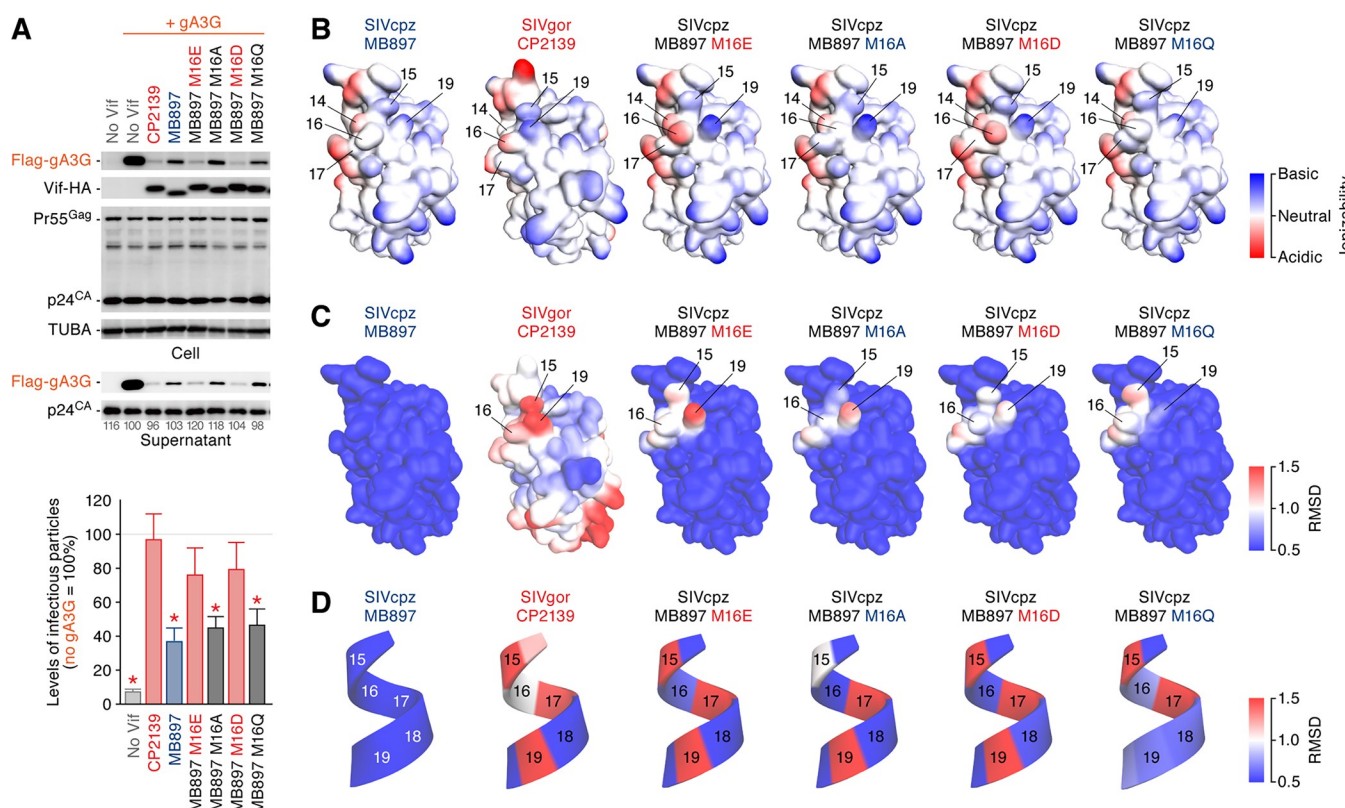

**Fig 3. Importance of acidic residues at position 16 of SIVcpz*Ptt* to counteract gA3G.** (A) Importance of acidic residue at position 16 of SIVcpz*Ptt* to counteract gA3G. HEK293T cells were co-transfected with pNL4-3Δ*vif* (500 ng) and the expression plasmids for gA3G (50 ng) and the indicated Vif strains and derivatives (500 ng). Cells and supernatants were harvested at two days post-transfection and were used for Western blotting (top) and TZM-bl assay (bottom). For Western blotting, the input of cell lysate was standardized to TUBA, and representative results are shown. The band intensity of viral p24 was quantified and the intensity value of the gA3G expressing cells without Vif (second from the left) was set to 100%. For TZM-bl assay, the percentage of the value without gA3G is shown. The mean values of nine independent experiments ± SEM are shown. Statistically significant differences (*P* < 0.05) versus "CP2139" are shown by red asterisks. (B-D) Structure homology models of Vif. The homology models are constructed based on the crystal structure of HIV-1M Vif (PDB: 4N9F) [29]. (B) Ionizability of each amino acid residue. Basic and acidic residues are indicated in blue and red, respectively. (C and D) Structural comparison. The topological changes of respective residues comparing to SIVcpz*Ptt* MB897 Vif are indicated based on the RMSD values of heavy atom positions. The whole structure model (C) and the cartoon model of the residues between 15–19 (D) are respectively shown.

## A Vif F1 box has been lost in Vif proteins of SIVgor and HIV-1OP

To neutralize hA3G and human A3F (hA3F), respectively, the G box (positioned at residues 40–44; also known as the YRHHY motif) and the F1 box (residues 14–17; also known as the DRMR motif) in HIV-1 Vif are indispensable [30, 31]. Since M16 is located within the F1 box, we next assessed the conservation of the amino acid residues in great ape lentiviral Vifs. As shown in **Fig 4A**, the Vif G box is highly conserved in all great ape lentiviruses. The importance of this motif is further evidenced by the fact that the Vif mutants that possess five alanines in residues 40–44 (correspond to the G box) of SIVgor and SIVcpz (the G/5A mutants) are unable to induce the degradation of gA3G (**Fig 4B**). In sharp contrast, it was intriguing that the F1 box is highly conserved in HIV-1M, HIV-1N, SIVcpz*Ptt* and SIVcpz*Pts*, whereas it is not conserved in HIV-1O, HIV-1P, and SIVgor (**Fig 4A**). Together with the phylogenetic analysis (**Fig 1A**), these findings show that the Vif F1 box has been lost in SIVgor and related HIV-1 groups.

## SIVgor Vif induces gorilla A3F degradation independently of F1 and F3 boxes

The absence of the Vif F1 box in SIVgor raises two possibilities: (i) SIVgor Vif is unable to counteract gorilla A3F (gA3F); or (ii) SIVgor Vif counteracts gA3F independently of the F1

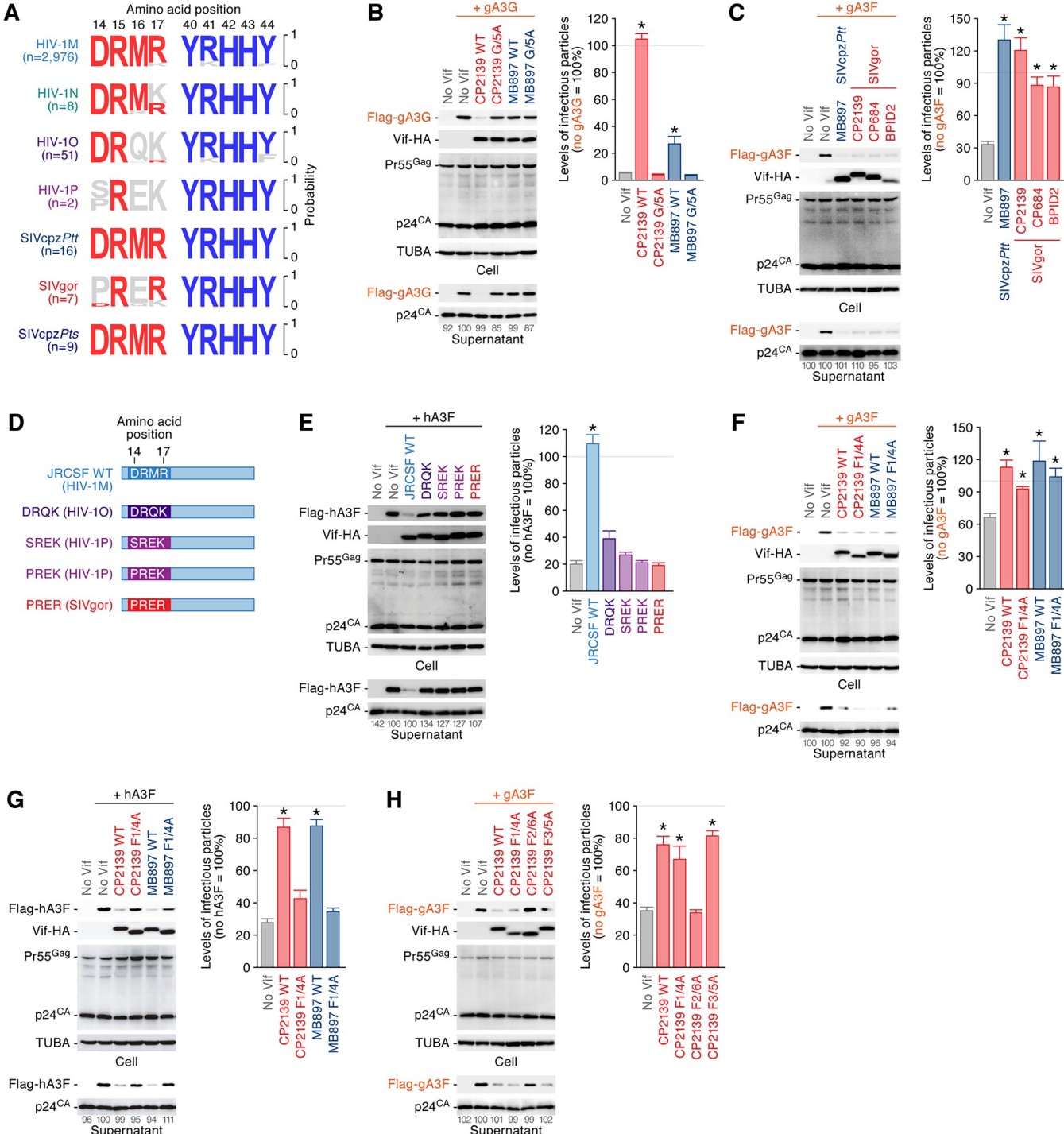

**Fig 4. Counteraction of antiviral gA3F by SIVgor Vif independently of the F1 and F3 boxes.** (A) Loss of F1 box in SIVgor, HIV-1O and HIV-1P Vif. Logo plots of the Vif residues between 14–17 (corresponding to the F1 box in HIV-1M; left) and the residues between 40–44 (corresponding to the G box in HIV-1M; right) in HIV-1MNOP, SIVcpz and SIVgor are shown. The number in parenthesis (n) indicates the number of viral sequences used in this analysis. (B and C) Counteracting activity against gA3G and gA3F. HEK293T cells were co-transfected with pNL4-3Δ*vif* (500 ng) and the expression plasmids for either gA3G (50 ng, B) or gA3F (200 ng, C) and indicated Vif (500 ng). (D) Scheme of HIV-1M JRCSF Vif derivatives used in this study. (E) Counteracting activity against hA3F. HEK293T cells were co-transfected with pNL4-3Δ*vif* (500 ng) and the expression plasmids for hA3F (50 ng) and indicated HIV-1M JRCSF Vif derivatives (500 ng). (F-H) Counteracting activity of SIVgor and SIVcpz Vif against great ape A3F. HEK293T cells were co-transfected with pNL4-3Δ*vif* (500 ng) and the expression plasmids for either gA3F (F, 200 ng) or hA3F (G, 50 ng) and indicated Vif (500 ng). In B, C, and E-H, cells and supernatants were harvested at two days post-transfection and were used for Western blotting and TZM-bl assay. For Western blotting, the input of cell lysate was standardized to

TUBA, and representative results are shown. The band intensity of viral p24 was quantified and the intensity value of the A3 expressing cells without Vif (second from the left) was set to 100%. For TZM-bl assay, the percentage of the value without A3 is shown. The mean values of three independent experiments ± SEM are shown, and statistically significant differences ($P < 0.05$) versus "no Vif" are shown by asterisks.

box. To distinguish between these possibilities, an expression plasmid for gA3F was co-transfected together with the Vif expression plasmids and an IMC of *vif*-deleted HIV-1. As shown in **Fig 4C** and **S4 Fig**, the Vif proteins including three strains of SIVgor (CP2139, CP684 and BPID2), five strains of SIVcpz*Ptt* (MB897, CAM3, DP943, MT145 and LB7) and the M16E mutants of SIVcpz*Ptt* Vif counteracted gA3F. These findings suggest that the antiviral effect of gA3F is neutralized by SIVcpz*Ptt* Vif and cannot be a barrier restricting cross-species transmission of SIVcpz*Ptt* to gorilla.

To assess the possibility that the residues 14–17 in HIV-1O, HIV-1P and SIVgor can be compensatory for the F1 box (DRMR motif) of HIV-1M, we constructed chimeric mutants based on HIV-1M strain JRCSF Vif, which possess the amino acid residues derived from HIV-1O, HIV-1P and SIVgor: DRQK (from HIV-1O), SREK (from HIV-1P), PREK (from HIV-1P) and PRER (from SIVgor) (**Fig 4D**). Although parental HIV-1 JRCSF Vif counteracted hA3F, these chimeric mutants did not (**Fig 4E**). To further assess whether SIVgor Vif can induce the degradation of gA3F independently of residues 14–17 (correspond to the F1 box), we generated the Vif mutants with alanines at these positions (the F1/4A mutants). These mutational analyses showed that the Vif F1/4A mutants of SIVcpz*Ptt* and SIVgor are still able to counteract gA3F (**Fig 4F**). Also, the M16E mutants of the 5 strains of SIVcpz*Ptt* Vifs degraded gA3F (**S4 Fig**). These findings raise a possibility that the importance of the F1 box is dependent on A3F. As shown in **Fig 4G**, we then used hA3F and found that SIVgor WT Vif, as well as SIVcpz*Ptt* Vif, counteract hA3F, while their F1/4A mutants do not. Collectively, these findings suggest that gA3F is neutralized by SIVcpz*Ptt* and SIVgor Vif independently of the F1 box, and the importance of the Vif F1 box is dependent on the host species of A3F.

Previous studies have revealed that the HIV-1M Vif regions other than the F1 box are crucial to degrade hA3F: the F2 box (residues 74–79, also known as the TGERxW motif) and the F3 box (residues 171–175, also known as the EDRWN motif) [32–34]. As shown in **S5 Fig**, the conservation levels of these motifs were different among viral clades. To investigate how SIVgor Vif degrades gA3F independently of the F1 box, we constructed the SIVgor Vif mutants with alanines at F2 and F3 boxes (the F2/6A and F3/5A mutants). Although the F3 box was dispensable for SIVgor Vif to degrade gA3F, the F2/6A mutant was unable to neutralize gA3F (**Fig 4H**). Altogether, these findings suggest that the F2 box of SIVgor Vif plays a pivotal role in neutralizing gA3F.

## Discussion

In the present study, we demonstrate that hA3G is counteracted by all great ape lentiviral Vifs tested. These results are consistent with previous observations that SIVcpz [35] and SIVgor [8] are able to replicate in *ex vivo* human CD4$^+$ T-cell culture. Moreover, SIVcpz efficiently expands in hematopoietic stem cell-transplanted humanized mouse models [24, 36]. These observations suggest that hA3G does not pose a barrier impeding SIV transmission from chimpanzees and gorillas to humans. As a notable exception, the human *A3H* gene is polymorphic and a haplotype, the human A3H haplotype II is resistant to SIVcpz and SIVgor Vif [25, 26]. These observations suggest that human A3H haplotype II may have hampered zoonotic transmission of SIVcpz from chimpanzees to humans. Notably, however, it remained unclear how passing through gorilla affects the evolution of great ape lentiviruses, particularly in terms of the relationship between lentiviral Vif and host A3. Here, we show that SIVcpz*Ptt* Vif is proficient to neutralize antiviral gA3F but is defective in efficiently triggering the degradation of

gA3G and thereby antagonizing the gA3G-mediated antiviral effect. We demonstrate that an amino acid substitution at position 16 of SIVcpz*Ptt* Vif is sufficient to acquire the ability to counteract gA3G. Together with our findings that antiviral activity of gA3G is relatively higher than those of other gorilla A3 proteins tested (A3D, A3F and A3H), it would be reasonable to speculate that gA3G restriction activity was a major hurdle for the transmission of SIVcpz*Ptt* from chimpanzees to gorillas. To our knowledge, this is the first report providing evidence that a great ape A3G protein plays a role in restricting the leap of a great ape lentivirus between different host species and how this hurdle was overcome.

Although the M16E substitution enabled SIVcpz*Ptt* Vif to neutralize gA3G efficiently, the reverse substitution in SIVgor Vif (i.e. E16M) did not cause a loss in counteraction. The reason for the discrepancy on the levels of gA3G between cells and virions remains unclear, but similar results have been reported in prior studies [37–39]. These observations imply that there may be a not-yet-identified factor(s) associated with the efficacy of A3 incorporation into the released virions, which is independent of Vif's ability to degrade A3 proteins in the cells. Thus, further investigations will be needed to fully elucidate the molecular mechanism of gA3G neutralization by SIVgor Vif. Collectively, our findings suggest that the acquisition of M16E mutation in Vif was a major step for SIVcpz to neutralize gA3G, but some additional mutations, which can fortify and/or compensate the ability of SIVgor Vif to degrade gA3G, may be needed during the adaptation to gorilla.

A key result is showing that a single amino acid substitution at position 16 (M16E) endows SIVcpz*Ptt* Vif with gA3G degradation activity. Structure-based models suggested that introduction of an acidic residue at position 16 may alter the side chain angles of adjacent basic residues at positions 15 and 19 and thereby reconfigures the Vif structure for efficient gA3G counteraction. Interestingly, in terms of the structural interaction between Vif and A3G, Letko *et al*. have provided a co-structure model of HIV-1 Vif and hA3G suggesting that the amino acid residues located at positions 14–17, 19 and 22 of Vif structurally interact with residues 125–130 of hA3G and that the electric surface charge of the respective domains determines Vif-A3G interaction [28]. Taken together with our findings, the acquisition of an acidic residue (glutamic acid or aspartic acid) at position 16 of SIVcpz*Ptt* Vif and subsequent repositioning of two basic residues at positions 15 and 19 confer the ability to counteract gA3G. Therefore, the electrostatic interactions between these residues are likely to be crucial for the functional interaction between Vif and gA3G. Additionally, it should be noted that this Vif region overlaps with the F1 box at positions 14–17, which is important for hA3F counteraction by HIV-1 Vif [30, 31]. However, it was surprising that SIVgor Vif counteracted gA3F in an F1 box-independent manner. To counteract gA3G, the structure of this region is reconfigured by acquiring acidic residue at position 16. Although it remains unclear how SIVgor Vif counteracts gA3F, our results suggest that the acquisition of an acidic residue at position 16 triggered the unique evolution of SIVgor Vif to counteract gA3F independently of the DRMR motif. Notably, HIV-1 groups O and P, the direct descendants of SIVgor [7], also lack a DRMR motif. In addition to this unique evolution of great ape lentiviral Vif in gorillas, previous studies suggest that other viral antagonists, Vpu and Nef, have uniquely evolved in gorillas: HIV-1 groups M and N switched from Nef- to Vpu-mediated counteraction of tetherin after zoonotic transmission of SIVcpz*Ptt* to humans [reviewed in [9, 17]]. In contrast, Kluge et al. revealed that Nef, but not Vpu, gained the ability to counteract human tetherin in case of HIV-1 group O [40]. Since HIV-1 group O emerged from SIVgor [7], this finding is another example of the unique evolution of lentiviral genes in gorillas.

Although the F1 box in Vif is dispensable for gA3F degradation, we showed that the other motif, F2 box, is essential. Regarding this, Richards et al. have shown that the F1 box of HIV-1 Vif may indirectly participate in hA3F [41]. Although the amino acid residues at positions 74–

79, which corresponds to the F2 box of HIV-1 group M Vif, are also different among great ape lentiviruses including SIVgor, our data suggest that the amino acid residues positioned at this region are important to maintain the electrostatic interaction between Vif and A3F.

In summary, we shed light on the evolutionary interplay between lentiviral Vif and host A3 in great apes and demonstrate that the interaction between Vif and A3 in gorillas has uniquely affected the evolutionary trajectory of great ape lentiviruses and therefore also the evolution of HIV-1 groups O and P. Furthermore, we provide functional and structural insights into how SIVcpz*Ptt* Vif overcame the species barrier mediated by gA3G. Thus, our studies provide important insights into the adaptive processes of great ape lentiviruses and will help to understand genetic and phenotypic differences of HIV-1 groups MN and OP that resulted from zoonotic transmissions from chimpanzees and gorillas, respectively.

## Materials and methods

### Cells

HEK293T cells (a human embryonic kidney cell line; ATCC CRL-1573), *CBFB* knock-down (KD) HEK293 cells [42] and TZM-bl cells (obtained through the NIH AIDS Research and Reference Reagent Program catalog number #8219) were maintained in Dulbecco's modified Eagle's medium (Sigma) containing FCS and antibiotics. The CD4$^+$ T cell line SupT11 expressing CCR5 (SupT11-CCR5) was prepared as reported previously [42]. To prepare the SupT11-CCR5 cells stably expressing gA3G (SupT11-CCR5-gA3G), a gA3G expression plasmid was electroporated into SupT11-CCR5 cells using the Neon Transfection system (Thermo Fisher Scientific). The transfected cells were selected with Puromycin (InvivoGen; 0.5 ng/ml) and the expression of gA3G in the selected cell clone was analyzed by Western blotting using anti-hA3G antibody (see below). Parental SupT11-CCR5 and SupT11-CCR5-gA3G cells were maintained in RPMI1640 (Sigma) containing FCS and antibiotics.

### Viruses

HEK293T cells and *CBFB* KD HEK293 cells [27] were transfected using PEI Max (Polysciences) according to the manufacturer's protocol. Basically, the expression plasmids for flag-tagged great ape A3 proteins were co-transfected with pNL4-3Δ*vif*, an infectious molecular clone of *vif*-deleted HIV-1M strain NL4-3 [30], and HA-tagged Vif expression plasmid into the cells. At two days post-transfection, the culture supernatants and transfected cells were harvested and were respectively used for TZM-bl assay and Western blotting as described below. For the preparation of infectious viruses, the IMCs (1,000 ng) were transfected into HEK293T cells. At two days post-transfection, the culture supernatants were harvested, centrifuged, and then filtered through a 0.45-μm-pore-size filter. To titrate virus infectivity, TZM-bl assay was performed as described below.

### Molecular phylogenetic analysis

The *vif* open reading frame (ORF) sequences (listed in **S1 Table**) were extracted from Los Alamos National Laboratory HIV sequence database (https://www.hiv.lanl.gov/components/sequence/HIV/search/search.html) and aligned by L-INS-I program in the MAFFT version 7.205 [43]. A maximum-likelihood phylogenetic tree (**Fig 1A**) was constructed using RAxML-NG program [44] applying general time-reversible model with gamma-distributed rate variation among sites and estimated portion of invariant sites. Pairwise p-distance (i.e., the number of base difference per site) between the *vif* genes of SIVcpz*Ptt* and SIVgor was calculated with MEGA7 [45], and the mean and standard deviation values were then calculated.

The logo plot of Vif amino acid sequence is constructed using WebLogo 3 (http://weblogo.threeplusone.com) and the residues at positions 14–17 (F1 box), 40–44 (G box), 74–79 (F2 box), and 171–175 (F3 box) are shown in **Fig 4A and S5 Fig**.

## Expression plasmids

To construct the expression plasmids for flag-tagged great ape A3s, pcDNA3.1 (Thermo Fisher Scientific) was used as a backbone. The expression plasmids for flag-tagged human A3 proteins were used in our previous studies [30, 46]. To construct the expression plasmids for flag-tagged A3 proteins of chimpanzee and gorilla, the ORFs of chimpanzee A3D (JN247642), chimpanzee A3F (XM_525658), cA3G (NM_001009001), chimpanzee A3H (EU861357), gorilla A3D (JN247649), gA3F (JN247640), gA3G (AY639868) and gorilla A3H (EU861358) were synthesized by the GeneArt gene synthesis service (Thermo Fisher Scientific). The obtained DNA fragments were inserted into EcoRV-NotI or EcoRI-XhoI sites of pcDNA3.1. To construct the expression plasmids for HA-tagged lentiviral Vifs, pDON-AI (Takara) was used as a backbone. The expression plasmid for the HA-tagged Vifs of HIV-1M NL4-3 (M19921), SIVcpz*Ptt* MB897 (JN835461) and SIVcpz*Ptt* MT145 (JN835462) were used in our previous study [24]. The HA-tagged Vif ORFs of HIV-1M JRCSF (M38429), HIV-1N DJO0131 (AY532635), HIV-1O BCF183, HIV-1O RBF206, HIV-1P RBF168 and SIVgor CP2139 (FJ424866) were prepared by PCR using their IMCs as the templates and primers listed in **S2 Table**. The HA-tagged Vif ORFs of HIV-1P U14788 (HQ179987), SIVcpz*Ptt* CAM3 (AF115393), SIVcpz*Ptt* DP943 (EF535993), SIVcpz*Ptt* LB7 (DQ373064), SIVgor CP684 (FJ424871) and SIVgor BPID2 (KP004994) were synthesized by the GeneArt gene synthesis service. The obtained DNA fragments were inserted into BamHI-SalI site of pDON-AI. The Vif mutants of SIVcpz*Ptt* MB897 and other strains were prepared by the GeneArt site-directed mutagenesis system using the primers listed in **S3 Table**. The IMCs of SIVcpz*Ptt* MB897 derivatives, M16E and E2X (*vif*-deleted) were generated by mutagenesis/overlap extension PCR using the primers listed in **S4 Table**. The HA-tagged JRCSF Vif derivatives (**Fig 4D and 4E**) were prepared by PCR using the HA-tagged JRCSF Vif expression plasmid as the template and primers listed in **S5 Table**. The expression plasmid for CBF-β was used in the previous study [27]. Nucleotide sequences were determined by a DNA sequencing service (Fasmac), and the sequence data were analyzed by Sequencher v5.1 software (Gene Codes Corporation).

## Western blotting

Western blotting was performed as described [30, 36] using the following antibodies: anti-HA mouse monoclonal antibody (clone 3F10; Roche), anti-Flag mouse monoclonal antibody (clone M2; Sigma-Aldrich), anti-p24 goat antiserum (ViroStat), anti-alpha-tubulin mouse monoclonal antibody (TUBA; clone DM1A; Sigma-Aldrich); anti-hA3G rabbit antiserum (the NIH AIDS Research and Reference Reagent Program catalog number #10201); and anti-CBF-β mouse monoclonal antibody (sc-56751; Santa Cruz). For Western blotting of viral particles, 370 μl of viral supernatant was ultracentrifuged at 100,000 × g for 1 h at 4°C using a TL-100 instrument (Beckman), and the pellet was lysed with 1 × SDS buffer. Transfected cells were lysed with RIPA buffer (25 mM HEPES [pH 7.4], 50 mM NaCl, 1 mM MgCl₂, 50 μM ZnCl₂, 10% glycerol, 1% Triton X-100) containing a protease inhibitor cocktail (Roche).

## TZM-bl reporter assay

TZM-bl assay was performed as described [30, 36]. Briefly, 10 μl of virus supernatant was inoculated into TZM-bl cells in 96-well plate (Nunc), and the β-galactosidase activity was measured

by using the Galacto-Star mammalian reporter gene assay system (Thermo Fisher Scientific) and a 2030 ARVO X multi-label counter instrument (PerkinElmer) according to the manufacturers' procedure. The relative infectivity was determined by relative light unit of this assay.

## Multi-round virus infection

The virus supernatant of SIVcpz*Ptt* MB897 WT, M16E and E2X (*vif*-deleted) derivatives were inoculated into SupT11-CCR5 (parental) and SupT11-CCR5-gA3G cells at multiplicity of infection (MOI) 0.1. The culture supernatant was routinely harvested and the amount of released viruses was measured by TZM-bl assay (**Fig 2G and 2H**).

## Construction and comparison of protein structure homology models

All protein structural analyses were performed using Discovery Studio v4.1 (Dassault Systèmes BIOVIA). First, the template crystal structure of HIV-1M Vif (PDB ID: 4N9F, chain v, 27) was selected using SIVgor CP2139 Vif and SIVcpz*Ptt* MB897 Vif amino acid sequences, then the sequences were aligned to the template using sequence alignment tool in Discovery Studio. Then, 100 homology models were generated for each SIV Vif using Build Homology Model protocol MODELLER v9.17 [47]. Evaluation of the homology models were performed using PDF total scores and DOPE scores and the best model for each SIV Vif was selected. After refining the models using protein refinement tools, SIVcpz*Ptt* MB897 Vif derivatives (M16E, M16A, M16D and M16Q) were generated based on the model of SIVcpz*Ptt* MB897 Vif using Build Mutant protocol in Discovery Studio. The root mean square deviation (RMSD) values between SIVcpz*Ptt* MB897 Vif and its derivatives and SIVgor CP2139 Vif were calculated on the heavy atom positions using RMSD calculation tool in Discovery Studio.

## Quantification and statistical analysis

The band intensity of viral p24 was quantified using ImageJ software (https://imagej.nih.gov/ij/index.html). Data analyses were performed using GraphPad Prism software. The data are presented as averages ± SEM. Statistically significant differences were determined by Student's *t* test. Statistical details can be found directly in the figures or in the corresponding figure legends.

## Supporting information

**S1 Fig. Antiviral activity of great ape A3D, A3F and A3H.** HEK293T cells were co-transfected with pNL4-3Δ*vif* (500 ng) and the different amounts of expression plasmids for great ape A3D (A), A3F (B) and A3H (C) (0, 50, 100, and 200 ng; the plasmid amount was normalized by empty vector). Cells and supernatants were harvested at two days post-transfection and were used for Western blotting (left) and TZM-bl assay (right). For Western blotting, the band intensity of viral p24 was quantified and the intensity value of the leftmost lane was set to 100%. For TZM-bl assay, the infectivity value without A3 was set to 100%.
(TIF)

**S2 Fig. Amino acid alignment of the SIVcpz*Ptt* MB897 Vif derivatives used in this study.** The scheme of respective mutants is also shown in **Fig 2A**.
(TIF)

**S3 Fig. Counteracting activity of the E16M mutant of SIVgor Vif against gA3G.** HEK293T cells were co-transfected with pNL4-3Δ*vif* (500 ng) and the expression plasmids for gA3G (50 ng) and indicated Vif (500 ng). Cells and supernatants were harvested at two days post-

transfection and were used for Western blotting (left) and TZM-bl assay (right). For Western blotting, the input of cell lysate was standardized to TUBA, and representative results are shown. The band intensity of viral p24 was quantified and the intensity value of the gA3G expressing cells without Vif (second from the left) was set to 100%. For TZM-bl assay, the percentage of the value without gA3G is shown. The mean values of three independent experiments ± SEM are shown.
(TIF)

**S4 Fig. Counteracting activity of SIVcpz*Ptt* Vif and the M16E mutant against gA3F.**
HEK293T cells were co-transfected with pNL4-3Δ*vif* (500 ng) and the expression plasmids for gA3F (200 ng) and indicated Vif (500 ng). Cells and supernatants were harvested at two days post-transfection and were used for Western blotting (left) and TZM-bl assay (right). For Western blotting, the input of cell lysate was standardized to TUBA, and representative results are shown. The band intensity of viral p24 was quantified and the intensity value of the gA3F expressing cells without Vif (second from the left) was set to 100%. For TZM-bl assay, the percentage of the value without gA3F is shown. The mean values of three independent experiments ± SEM are shown, and statistically significant differences ($P < 0.05$) versus "no Vif" are shown by asterisks.
(TIF)

**S5 Fig. Conservation of F2 and F3 boxes in SIVgor, HIV-1O and HIV-1P Vif.** Logo plots of the Vif residues between 74–79 (corresponding to the F2 box in HIV-1M; left) and the residues between 171–175 (corresponding to the F3 box in HIV-1M; right) in HIV-1MNOP, SIVcpz and SIVgor are shown. The number in parenthesis (n) indicates the number of viral sequences used in this analysis.
(TIF)

**S1 Table. Accession numbers of the viruses used in this study.** A full list of the accession numbers of viral strains used in **Fig 1A**.
(XLSX)

**S2 Table. Primers for the construction of Vif expression plasmids.** A full list of the primers used for the construction of Vif expression plasmids.
(XLSX)

**S3 Table. Primers for the construction of SIVcpz*Ptt* Vif derivatives.** A full list of the primers used for the construction of SIVcpz*Ptt* Vif derivatives.
(XLSX)

**S4 Table. Primers for the construction of SIVcpz*Ptt* MB897 IMC derivatives.** A full list of the primers used for the construction of SIVcpz*Ptt* MBV897 IMC derivatives.
(XLSX)

**S5 Table. Primers for construction of HIV-1M JRCSF Vif derivatives.** A full list of the primers used for the construction of HIV-1M JRCSF Vif derivatives.
(XLSX)

## Acknowledgments

We would like to thank Beatrice H. Hahn and Frederic Bibollet-Ruche (University of Pennsylvania, USA), and Frank Kirchhoff and Daniel Sauter (Ulm University, Germany) for kindly providing the IMCs of some HIV-1, SIVcpz*Ptt* and SIVgor. We also thank Kotubu Misawa for dedicated support.

## Author Contributions

**Conceptualization:** Kei Sato.

**Data curation:** Mahoko Takahashi Ueda, So Nakagawa.

**Formal analysis:** Mahoko Takahashi Ueda, Jumpei Ito, So Nakagawa.

**Funding acquisition:** Yusuke Nakano, So Nakagawa, Terumasa Ikeda, Yoshio Koyanagi, Reuben S. Harris, Kei Sato.

**Investigation:** Yusuke Nakano, Keisuke Yamamoto, Mahoko Takahashi Ueda, Andrew Soper, Yoriyuki Konno, Izumi Kimura, Keiya Uriu, Ryuichi Kumata, Hirofumi Aso, Naoko Misawa, Shumpei Nagaoka, Soma Shimizu, Keito Mitsumune, Yusuke Kosugi, Guillermo Juarez-Fernandez, Jumpei Ito, Terumasa Ikeda.

**Methodology:** Yusuke Nakano, Keisuke Yamamoto, Naoko Misawa, Jumpei Ito.

**Project administration:** Kei Sato.

**Resources:** So Nakagawa, Terumasa Ikeda, Yoshio Koyanagi, Reuben S. Harris.

**Software:** Mahoko Takahashi Ueda, So Nakagawa.

**Supervision:** Yoshio Koyanagi, Reuben S. Harris, Kei Sato.

**Validation:** Yusuke Nakano, Keisuke Yamamoto.

**Visualization:** Yusuke Nakano, Keisuke Yamamoto, Mahoko Takahashi Ueda, Naoko Misawa, Kei Sato.

**Writing – original draft:** Kei Sato.

**Writing – review & editing:** Yusuke Nakano, Keisuke Yamamoto, Mahoko Takahashi Ueda, Andrew Soper, Yoriyuki Konno, Izumi Kimura, Keiya Uriu, Ryuichi Kumata, Hirofumi Aso, Naoko Misawa, Shumpei Nagaoka, Soma Shimizu, Keito Mitsumune, Yusuke Kosugi, Guillermo Juarez-Fernandez, Jumpei Ito, So Nakagawa, Terumasa Ikeda, Yoshio Koyanagi, Reuben S. Harris, Kei Sato.

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
