## [Decision Letter · Decision Letter 0]

25 Jun 2020

Dear Dr. Sato,

Thank you very much for submitting your manuscript "A role for gorilla APOBEC3G in shaping lentivirus evolution including transmission to humans" for consideration at PLOS Pathogens. As with all papers reviewed by the journal, your manuscript was reviewed by members of the editorial board and by independent reviewers. In light of the reviews (below this email), we would like to invite the resubmission of a significantly-revised version that takes into account the reviewers' comments. We would like to note that obtaining reviews from another source can improve the manuscript and facilitate publication but does not entirely satisfy the need for review by independent reviewers chosen by the journal. In this case, we obtained two additional reviews, which are attached. Reviewer 2 felt you had not proven your point and recommended rejection. This seems to us a too severe viewpoint. Reviewer 1, in contrast, noted that you had not effectively responded to the first point made in the reviews you obtained, which clearly stated that experiments using stocks of different viruses to compare infection rates must use stocks normalized based on p24Gag levels measured by ELISA, not by Western blot, which is only semi-quantitative. Therefore, prior to acceptance, all such experiments must be repeated using stocks normalized by ELISA.

We cannot make any decision about publication until we have seen the revised manuscript and your response to the reviewers' comments. Your revised manuscript is also likely to be sent to reviewers for further evaluation.

Sincerely,

Bryan R. Cullen

Associate Editor

PLOS Pathogens

Thomas Hope

Section Editor

PLOS Pathogens

Kasturi Haldar

Editor-in-Chief

PLOS Pathogens

orcid.org/0000-0001-5065-158X

Michael Malim

Editor-in-Chief

PLOS Pathogens

orcid.org/0000-0002-7699-2064

Reviewer's Responses to Questions

**Part I - Summary**

Reviewer #1: Nakano et al show that the lentivirus that passed from chimpanzees to gorillas to establish SIVgor adapted to the APOBEC3G and APOBEC3F genes found in gorillas. The authors map the this change to amino acid changes in Vif . This paper is a nice addition to the literature about how SIVs adapted to new hosts through cross species transmissions via the Vif-APOBEC3 antagonism. The present manuscript was already reviewed (I was not involved in the previous review). The authors have adequately responded to the previous reviewer comments with one exception described below.

Reviewer #2: This manuscript reports studies on the mechanism of why SIVcpzPtt Vif does not neutralize gorilla APOBEC3G (gA3G). Because g3G is neutralized by SIVgor Vif, chimeras between SIVcpzPtt and SIVgor Vif proteins were created to map the critical determinant. It was found that this species-specific activity is determined by an amino acid residue at position 16, where SIVcpzPtt and SIVgor Vif proteins have a Met (M16) or Glu (E16) residue, respectively. When a M16E mutation was introduced, SIVcpzPtt Vif became able to neutralize gA3G. Based on this observation, it was concluded that the Vif M16E mutation plays a key role in SIV transmission from chimpanzees to gorillas and then to humans, as described in Title, Abstract, Introduction, and Discussion. However, even though this residue determines SIV Vif activity to neutralize A3G, it does not necessarily mean it has played a role in SIV evolution and cross-species transmission. There is no any evidence provided in this manuscript that supports the central conclusion, which limits its significance of these studies.

**Part II – Major Issues: Key Experiments Required for Acceptance**

Reviewer #1: Major Comments:

1. I am concerned about the authors’ reply to Reviewer #1’s query about whether or not the virus stocks were normalized for p24 before infectivity assays. The authors reply that they were not, but instead point out that Western blots show the same amounts of p24gag. While this may be qualitatively true, much of the data show significance where there is a two-fold difference in infectivity (for example, Figure 4). The standardization of the virus stocks therefore becomes critical to show that variation in virus production does not account for this difference. While the Western blots look about the same, they need to be quantified. This is why most groups routinely standardize their virus stocks through more precise methods (p24 ELISA or RT activity, for example). The authors’ contention that APOBECs and Vif do not affect virus production is not completely true since transfection artifacts of over-expression do lead to changes in virus amounts and that needs to be accounted for. At the least, the authors need to quantify their Western blots and show this data

Reviewer #2: 1. A major concern is whether the Vif M16E mutation could occur naturally during SIV evolution. In Fig.2G, a SupT11-CCR5-gA3G cell line was used to propagate SIVcpzPtt. Can the M16E mutation be found in viruses recovered from the cell culture? If not, it would suggest that there are other factors that determine SIV cross-species transmission.

2. In addition to gA3G and gA3F, the activity of gA3D and gA3H should be also determined in order to justify the central role of gA3G in SIVcpzPtt transmission.

**Part III – Minor Issues: Editorial and Data Presentation Modifications**

Reviewer #1: (No Response)

Reviewer #2: In the rebuttal letter, it was shown that unlike the M16E mutation that enables SIVcpzPtt Vif to neutralize gA3G, an E16M mutation did not disable SIVgor Vif from neutralizing gA3G. These results should be included and fully discussed in the manuscript.

PLOS authors have the option to publish the peer review history of their article (what does this mean?). If published, this will include your full peer review and any attached files.

Reviewer #1: No

Reviewer #2: No
---

## [Editor Report · Decision Letter 1]

15 Jul 2020

Dear Dr. Sato,

We are pleased to inform you that your manuscript 'A role for gorilla APOBEC3G in shaping lentivirus evolution including transmission to humans' has been provisionally accepted for publication in PLOS Pathogens.

Best regards,

Bryan R. Cullen

Associate Editor

PLOS Pathogens

Thomas Hope

Section Editor

PLOS Pathogens

Kasturi Haldar

Editor-in-Chief

PLOS Pathogens

orcid.org/0000-0001-5065-158X

Michael Malim

Editor-in-Chief

PLOS Pathogens

orcid.org/0000-0002-7699-2064
---

## [Editor Report · Acceptance letter]

7 Aug 2020

Dear Dr. Sato,

We are delighted to inform you that your manuscript, "A role for gorilla APOBEC3G in shaping lentivirus evolution including transmission to humans," has been formally accepted for publication in PLOS Pathogens.

Best regards,

Kasturi Haldar

Editor-in-Chief

PLOS Pathogens

orcid.org/0000-0001-5065-158X

Michael Malim

Editor-in-Chief

PLOS Pathogens

orcid.org/0000-0002-7699-2064